

# Identification of potential glioma drug resistance target proteins based on ultra-performance liquid chromatography-mass spectrometry differential proteomics

Da Li, Jun Yan, Kang Li, Qingcheng Yang, Liping Bian, Bencheng Lin, Xiaohua Liu and Zhuge Xi

Tianjin Institute of Environmental and Operational Medicine, Tianjin, China

Corresponding authors
Xiaohua Liu,
liuxiaohua1992@sina.com
Zhuge Xi, zhugexi2003@sina.com

## ABSTRACT

In this study, to screen for candidate markers of temozolomide (TMZ) resistance in glioblastoma, we artificially established TMZ drug-resistant glioblastoma (GBM) cell lines, U251-TMZ and U87-TMZ. In the U251-TMZ and U87-TMZ cell lines, we screened and analyzed differentially expressed proteins using ultra-performance liquid chromatography-mass spectrometry (UPLC-MS) differential proteomics. Compared with the U251 and U87 control cell lines, 95 differential proteins were screened in the U251-TMZ and U87-TMZ cell lines, of which 28 proteins were upregulated and 67 proteins were down-regulated. Gene Ontology (GO) and Kyoto Encyclopedia of Genes and Genomes (KEGG) analyses of the co-upregulated proteins showed that most of the differentially expressed proteins were located in the cytoplasm and were significantly upregulated in the biological processes related to vesicular transport in the intimal system and inflammatory response mediated by myeloid leukocytes. Seven candidates were identified as potential GBM markers of TMZ resistance. Combined with existing research findings, our study supports that UAP1L1 and BCKDK are promising potential markers of TMZ resistance in GBM. This is important for further understanding the molecular mechanisms that drive the development and enhancement of TMZ resistance.

## INTRODUCTION

Glioblastoma (GBM) is one of the most common and recurrent primary malignant brain tumors. GBM accounts for approximately 46% of all diagnosed brain tumors. According to statistics, 2.7% of cancer-related deaths are caused by GBM (*Lyon et al., 2017*). The average survival time of GBM patients is only 12–15 months. Extensive studies have been performed to understand and treat GBM, but the results remain unsatisfactory. The short survival time is related to the anatomical location of the tumor, heterogeneity of GBM cells, and rapid growth of GBM (*Shergalis et al., 2018*). The incidence of GBM increases with age,

and the incidence in men is slightly higher than that in women (*Yang et al., 2019a*). It is listed as the third most common cause of cancer-related death in patients aged 15–34 years. Clinically, standard GBM treatment includes the maximum safety of surgical resection of tumors, radiotherapy, and adjuvant chemotherapy (*Delgado-Lopez & Corrales-Garcia, 2016*).

Since temozolomide (TMZ) was approved by the FDA in 2005, it has been widely used as a standard chemotherapy regimen for GBM diagnosis (*Hanif et al., 2017*; *Xu et al., 2020*). TMZ can cross the blood–brain barrier and can be administered orally. It is activated by conversion to the metabolite 5-(3-methyltriazen1-yl) imidazole-4-carboxamide (MTIC) at physiological pH values (*Stupp et al., 2014*). However, TMZ is also a key factor in drug resistance and recurrence. Due to the high heterogeneity and mutagenicity of GBM and extensive exposure to TMZ, >50% of GBM patients treated with TMZ do not respond to treatment (*Lee, 2016*). However, TMZ remains the key to the treatment of high-level gliomas. TMZ resistance is a significant issue that must be overcome for the successful treatment of GBM.

At present, proteomics technology can reflect the tumor phenotype more accurately, and its advantage is better than genome and transcriptome analyses (*Khalil, 2007*). The level of gene expression is not always related to the level of protein expression, and the process of post-translational repair cannot be detected (*Niclou, Fack & Rajcevic, 2010*). Because of the limited predictive markers of TMZ resistance (*Arora & Somasundaram, 2019*), anti-TMZ resistance has become more complex for GBM treatment (*Wick & Platten, 2014*). Therefore, it is urgent to identify reliable biomarkers that can be used to determine TMZ resistance.

In this study, two classic GBM cell lines, U251 and U87, were used to identify reliable biomarkers for TMZ resistance. Firstly, TMZ-resistant U251 and U87 GBM cell lines were constructed, and the protein expression profiles of U251-TMZ and U87-TMZ were compared using the non-quantitative labeling proteomic technique ultra-performance liquid chromatography-mass spectrometry (UPLC-MS). Twenty-seven upregulated differential proteins were screened, among which seven were identified according to the expression difference multiple and the credibility of Kyoto Encyclopedia of Genes and Genomes (KEGG) enrichment analysis. Bioinformatics analysis of differentially expressed proteins showed that the upregulated differentially expressed proteins were mainly related to energy metabolism, ribosome synthesis, and processing, and were enriched in endocytosis, carbon metabolism, and antibiotic synthesis. Our study summarizes the protein expression profiles in U251 and U87TMZ drug-resistant cell lines, which provides a basis for a better understanding of TMZ resistance in disease.

## MATERIALS & METHODS

### Materials

U251 and U87 cell lines were purchased from Tianjin Genink Biotechnology Co., Ltd. (Tianjin, China), DMEM and fetal bovine serum were purchased from Gibco (Waltham, MA, USA), and the Cell Counting Kit-8 was purchased from Dojindo Laboratories, Kumamoto, Japan.

## Screening of the U251 U87-TMZ cell line

Glioma cells in the logarithmic phase were added with 200 µM TMZ solution (diluted with DMEM complete medium). After 24 h, the medium was replaced, and 200 µM TMZ solution was added after 2 d of culture. When glioma cells proliferate and there is no significant change in cell morphology, glioma cells were treated with a higher concentration of TMZ according to the procedure described above. Repeat the above steps as the TMZ concentration gradually increases, such as 300, 400, 500 µM. to obtain TMZ resistant cell lines for 60 days (*Lee, 2016*). Viability curve of U87-TMZ and U251-TMZ for TMZ resistance compared with their parental cell lines were demonstrated in Fig. S1.

## Liquid-phase MS

The type of liquid chromatography was EASY-nLC1200 (Thermo Fisher Scientific) and the tandem mass spectrometer was Q Exactive HF (Thermo Fisher Scientific). The peptides were digested with phase A (0.1% formic acid) loaded onto a self-made Trap column (100 um × 2 cm C18 packing specification is 3umline 120A). Different gradients of phase B (80% acetonitrile, 0.1% formic acid, and 19.9% water) were used to elute the trap column. The eluted peptides were separated using an analytical column (150 um × 15 cm C18 packing specification, 1.9 um120A) to form a charged spray, which was detected using MS. The B phase gradient was 0 min, 7%; 8 min, 14%; 60 min,28%; 72 min, 42%; 73 min, 95%; 78 min, 95%; and 79 min, 30%, and the flow rate was 600 nL/min. Mass spectrum parameters: The first-order detector type was an orbital trap (Orbitrap) with a resolution of 120,000; Orbitrap scan range, 300–1,400 m/z; max injection time, 80 ms; and AGC Target, 3E6. The second stage adopted a data-dependent scanning mode and the top 20 precursor ions were selected for secondary acquisition in each scan. The secondary detector was the same Orbitrap; resolution, 15,000 at 120 m/z; maximum injection time, 19 ms; AGC Target, 2E4; scan range, 200–2,000 m/z; included charge state, 2–6; collision energy duration, 15 s; mass tolerance, ±7 ppm; isolation window, 1.6 m/z; activation type, HCD; and normalized collision energy, 27%.

## Data processing and statistical analysis

Microsoft Excel 2016 was used for preliminary data collation. We performed all statistical analyses using GraphPad Prism 8. We used unpaired t test for comparison of cell viability between cell lines. The results were expressed as means ± standard deviations, and $P < 0.05$ was considered significant. We considered $p < 0.05$ to indicate statistical significance *** $p < 0.001$ **** $p < 0.0001$.

MaxQuant software (version 1.6.1.0; https://www.maxquant.org/) was used to search the database of the original data collected using MS. Restriction endonuclease digestion method: trypsin/P, and a maximum of two missing sites were allowed. Variable modification: oxidation (M) and acetyl (protein N-term). Fixed modification: carbamidomethyl (C), a maximum of five variable modifications, was allowed for each peptide. The first-level quality deviation was 20 ppm. The second-level quality deviation was 20 ppm. The database was Uniprot's SWISS database (containing 20,431 annotated proteins, released2019.07). The minimum peptide length was seven amino acids, and the
maximum peptide mass was 5,000 Da. The false discovery rate (FDR) of spectrum and protein matching was set to 1% and the protein quantitative value was obtained using the iBAQ algorithm.

After the median normalization of the original quantitative value, log2 transformation was performed to ensure that it met the normal distribution criteria, and the quantitative value of each sample met the normal distribution criteria. Proteins that could be statistically analyzed were selected (quantification of at least three samples in one group) and the minimum value for vacancies worthy of interpolation was selected. The difference in protein was analyzed using the R software $t$-test and function, and the $p$ value was corrected using the Benjamini–Hochberg algorithm. Proteins with adjusted $P < 0.05$, and fold change > 2 were considered differentially expressed proteins. The upregulated differential proteins were analyzed using the Cluster Profiler Package for GO and KEGG pathway enrichment analyses. Protein-protein interaction (PPI) analysis (STRING database, version 11.0) was also performed.

## RESULTS

### Screening of TMZ-resistant cell lines

U251 and U87 glioma cell lines in the logarithmic growth phase were added to the 200 µM TMZ solution in the beginning and cultured for 2 d followed by the addition of TMZ solution. The above procedures were repeated until the proliferation and morphology of glioma cells did not change significantly at this TMZ concentration. Replace higher concentration of TMZ sequentially and repeat the abovementioned steps to construct TMZ-resistant U251-TMZ and U87-TMZ cell lines. In addition to observing the morphological changes in cells to confirm the state of drug resistance, we also determined the OD value of cell-counting kit-8 (CCK8). As shown in Figs. 1A and 1B, there was no significant difference in morphology and quantity between the control and + TMZ groups of the U251-TMZ and U87-TMZ cell lines. The OD values of CCK8 in the U251-TMZ and U87-TMZ cell lines were significantly higher than those of their respective control groups. Therefore, we successfully constructed and screened TMZ-resistant glioma U251 and U87 cell lines.

### Data quality control

To better analyze the differential protein expression between TMZ drug-resistant cell lines U251 and U87, the quantitative data were normalized to the median. A comparison of the results before and after normalization is shown in Fig. 2A. Normalized data were analyzed using Pearson's correlation coefficient (Fig. 2B). The correlation coefficients between the samples were higher than those between the groups, and there were significant differences in protein expression between the two groups. Additionally, the quantitative data were also analyzed by cluster analysis, and the results showed that each group of biological repeats was clustered together, indicating high similarity and good data repeatability (Fig. 2C). Analysis of the coefficient of variation of the normalized data (Fig. 2D) showed that the coefficient of variation was low, and the quantitative stability of the protein was high.

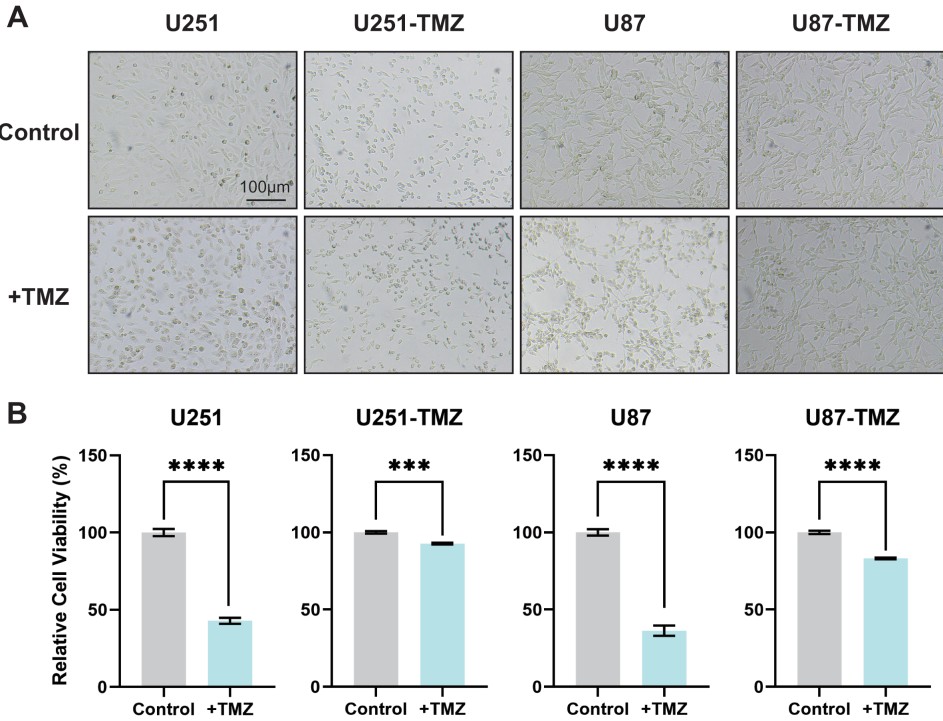

**Figure 1 Screening of TMZ-resistant glioma cell lines U251 and U87.** The culture status of U251-TMZ, U87-TMZ cells and control cell lines. (A) The relative cell viability of U251-TMZ, U87-TMZ cells and control cell lines, as measured by CCK-8 assay (B).

## Protein quantitative analysis

In this study, some of the 4,160 proteins detected were statistically available for analysis, including 3,308 in the U251 group and 3,621 in the U87 group. We screened the differentially expressed proteins in the four groups, particularly the upregulated proteins, and focused on screening and analyzing the differentially upregulated proteins in the U251-TMZ and U87-TMZ cell lines.

### Cross-comparison of differential proteins between the U251-TMZ and U87-TMZ cell lines

A total of 1,403 differential proteins were detected in the U251 group, of which 467 were upregulated and 936 were down-regulated in the U251-TMZ cell line. A total of 676 differentially expressed proteins were detected in the U87 group, of which U87-TMZ upregulated 386 proteins and down-regulated 290 (see Data S1). Because of the heterogeneity of GBM, we only screened 28 proteins upregulated from the upregulated proteins of the U251-TMZ and U87-TMZ cell lines (three repetitive samples), and screened 67 down-regulated proteins from their respective down-regulated proteins. A total of 95 differential proteins were screened, as shown in the differential protein clustering thermogram (Fig. 3A). Owing to the lack of statistical significance of the down-regulated proteins, we did not perform follow-up bioinformatics analysis of the down-regulated proteins, but focused on the 28 proteins that were upregulated. For each quantifiable

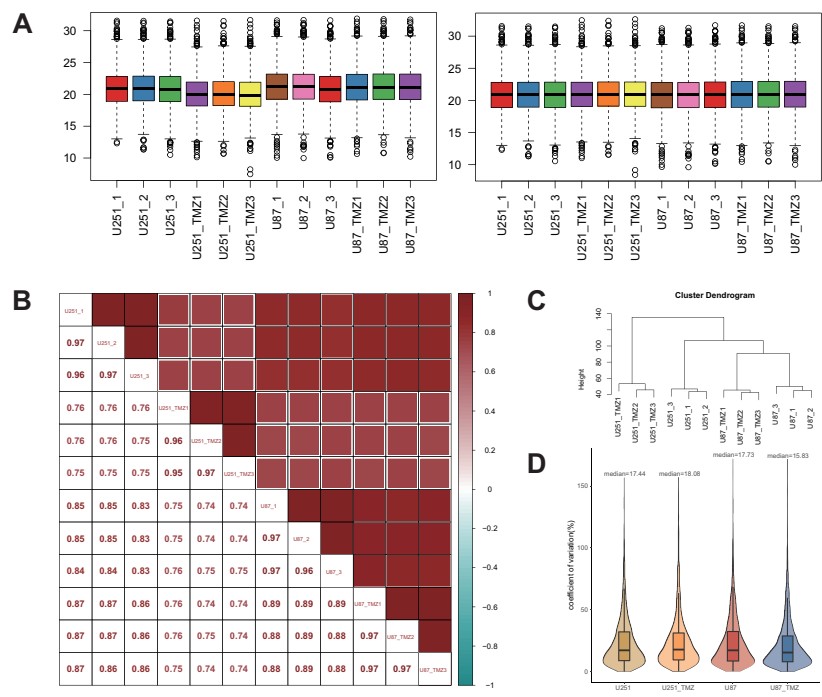

**Figure 2 Quantitative protein data quality control.** Before and after the median normalization (A left and right). The correlation between samples (B). The correlation between U251-1 and U251-2 is 0.97, the correlation between U251-1 and U251-3 is 0.96, the correlation between U251-2 and U251-3 is 0.97, the higher the correlation is, the closer the sample is. Cluster analysis of U251, U87, U251-TMZ and U87-TMZ samples (C). The coefficients of variation of U251, U87, U251-TMZ and U87-TMZ proteins were analyzed (D).

protein, the fold change was logarithmically calculated from a base of 2, and the *P* value was logarithmically calculated from a base of 10 to create a volcano plot (Fig. 3B).

## Functional annotation enrichment analysis of protein
### *GO and KEGG pathway analysis of the U251-TMZ and U87-TMZ cell lines*
A total of 28 proteins upregulated in the U251-TMZ and U87-TMZ cell lines were analyzed using GO and KEGG pathways, respectively. The possible pathways from biological process (BP), cellular component (CC), and molecular function (MF) were screened. As shown in Fig. 4A, these differentially expressed proteins were significantly upregulated in biological processes related to myeloid leukocyte activation, myeloid leukocyte-mediated immunity, leukocyte degranulation, and inflammation. Based on GO cell composition analysis, most of them were located in the cytoplasm and may be involved in vesicle transport in the intimal system. In addition, these proteins exhibited GO molecular functions of "catalytic activity" and "binding", such as aminopeptidase activity and coenzyme binding. These results suggest that the development of TMZ drug resistance may be related to intracellular immune regulation and energy metabolism (*Fu et al., 2021*; *St-Coeur et al., 2015*).

In addition, the enrichment of the 28 protein KEGG pathways upregulated in the U251-TMZ and U87-TMZ cell lines mainly included carbon metabolism, antibiotic biosynthesis,

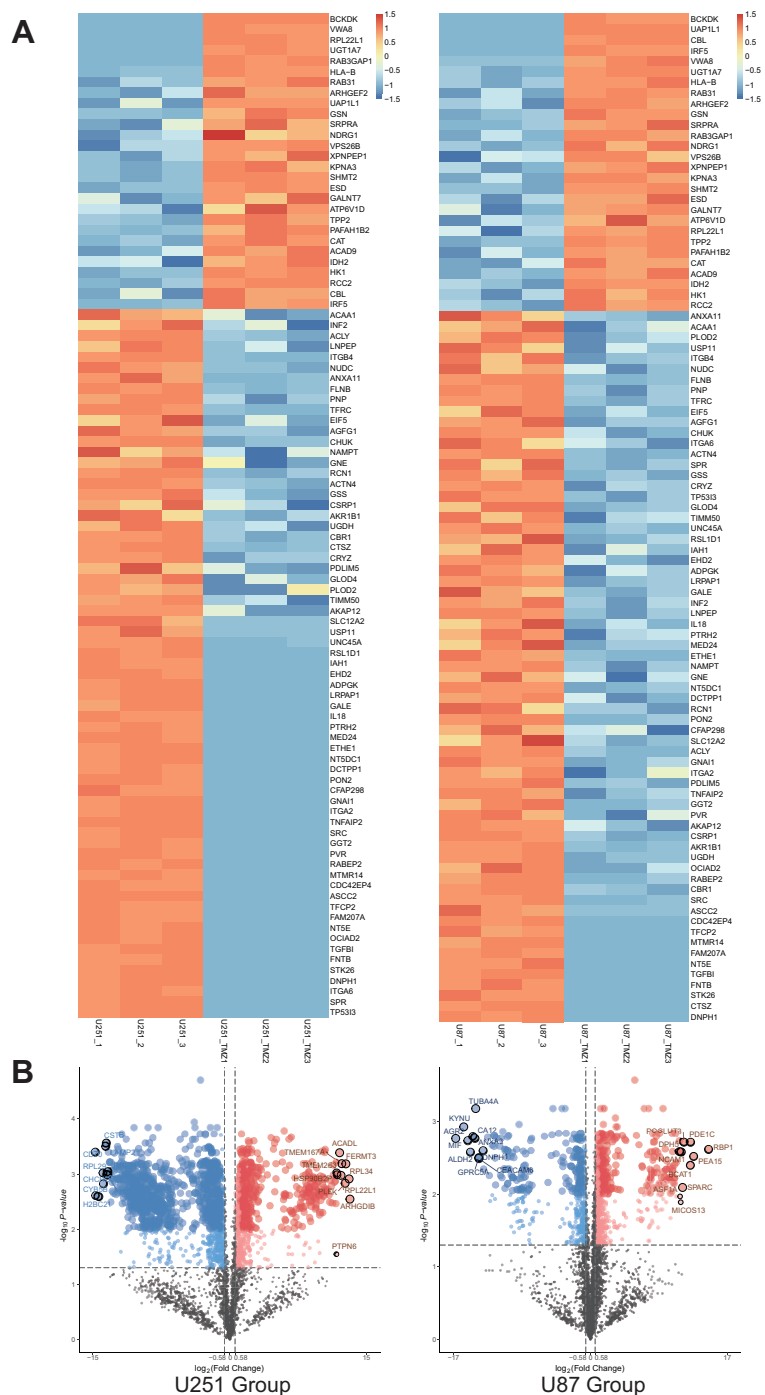

**Figure 3  Differential protein cross-comparison.** The differential proteins of U251-TMZ and U251 (left), U87-TMZ and U87 (right) were clustered respectively. U251-TMZ and U87-TMZ up-regulated 28 proteins and down-regulated 67 proteins (A). U251-TMZ and U251, U87-TMZ and U87 can be statistically analyzed protein volcano diagram (B).

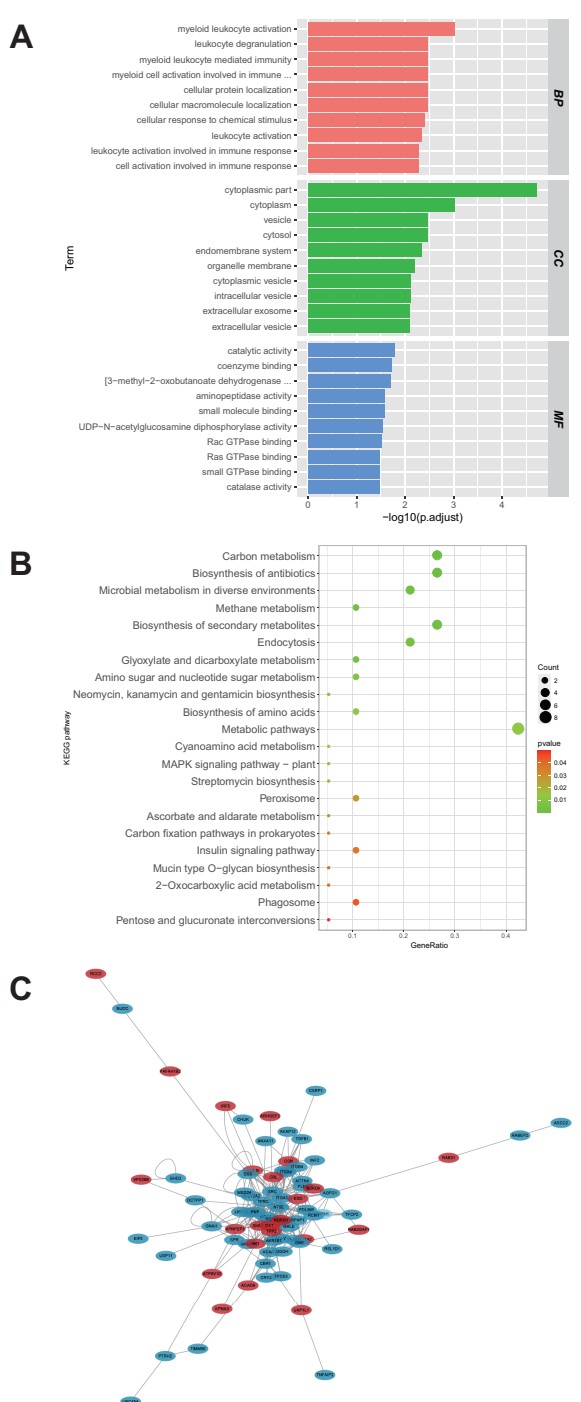

**Figure 4  Functional analysis of proteins up-regulated by U251-TMZ and U87-TMZ.** GO analysis of proteins up-regulated by U251-TMZ and U87-TMZ, including biological processes (Biological Process BP), cellular components (CC), and molecular function MF (A). KEGG signal pathway enrichment analysis of proteins up-regulated by U251-TMZ and U87-TMZ (B). Differential protein interaction network (C).

**Table 1  Potential markers for TMZ resistance in GZM.**

| Gene symbol | Gene ID | Description |
| --- | --- | --- |
| UGT1A7 | 54577 | UDP glucuronosyltransferase family 1 member A7 |
| BCKDK | 10295 | Branched chain keto acld dehydrogenase kinase |
| UAP1L1 | 91373 | UDP-N-acelylglucosamine pyrophosphorylase 1 like 1 |
| RAB3GAP1 | 22930 | RAB3 GTPase aclivating protein catalyic subunit 1 |
| WWA8 | 23078 | Von Willebrand factor A domain containing 8 |
| HL A-B | 3106 | Major histocompatblity complex, class I, B |
| VPS26B | 112936 | VPS26, retromer complex component B |

secondary metabolite biosynthesis, glyoxylic acid and dicarboxylic acid metabolism, and endocytosis (Fig. 4B).

## Differential protein interaction network

In the interaction network of proteins upregulated in the U251-TMZ and U87-TMZ cell lines (Fig. 4C), the changes in these proteins are related to immune regulation and vesicular transport, as well as to energy metabolism, such as the mitochondrial stress response.

## Potential GBM markers of TMZ resistance

In the MS analysis of differentially expressed proteins in the U251/U251-TMZ and U87/U87-TMZ cell lines, 28 proteins were upregulated. Through manual screening, seven proteins with the highest differential expression in the two groups were identified: UGT1A7, BCKDK, UAP1L1, RAB3GAP1, VWA8, HLA-B, and VPS26B (Table 1). At present, there is no research on the relationship between these seven proteins and TMZ resistance. Among these proteins, only UAP1L1 was related to GBM studies, and the expression of UAP1L1 in glioma tissues was significantly upregulated. Increased expression of UAP1L1 is associated with higher tumor grade and poor prognosis. UAP1L1 may promote the proliferation of glioma cells by regulating glycosylation of key proteins (Yang et al., 2021b).

Another noteworthy protein, BCKDK, has recently been found to regulate the catabolism of branched chain amino acids (BCAA) and enhance MEK/ERK signal transduction, which is a key pathway driving the growth and proliferation of cancer cells (Xue et al., 2017). Among the remaining five candidate proteins, it has been reported that UGT1A7 may affect the progression of pancreatic cancer (Yilmaz et al., 2015). Rab3GAP1 was first isolated from the synaptic soluble section of the rat brain as an Rab3A-binding protein, and it exhibits GAP activity against Rab3 (Handley & Aligianis, 2012). Rab3GAP deficiency leads to neurological disorders (Muller et al., 2011). VWA8 is expressed in organs with high energy requirements, including those with a high density of mitochondria (Luo et al., 2017). Upregulation of VWA8 expression was negatively correlated with brain metastasis associated with breast cancer (Yuan, Wang & Cheng, 2018). HLA is necessary for T-cell activation and is associated with multiple sclerosis (Hollenbach et al., 2016). VPS26 is one of the components of the cargo recognition core of the reverse transcriptase complex, which is significantly decreased in the hippocampus of patients with Alzheimer's disease. Changes in reverse transcriptase composition are related to an increased risk of Alzheimer's disease (Small et al., 2005; Vagnozzi & Pratico, 2019).

## Verification of potential markers of TMZ resistance

To verify the results of MS analysis of the U251/U251-TMZ and U87/U87-TMZ cell lines, we performed western blotting on seven selected markers: UGT1A7, BCKDK, UAP1L1, RAB3GAP1, VWA8, HLA-B, and VPS26B. The results showed that in the four cell lines, the above seven proteins were successfully upregulated in the U251-TMZ and U87-TMZ cell lines compared to the control groups (Fig. 5). Tet-On systems for doxycycline (Dox)-inducible gene expression were applied in our two resistant cell lines. The addition of Dox reduced the expression of these potential target genes. The cell growth was inhibited after the addition of Dox at Day 4 to reduce the target gene and was restored after removement of Dox at Day 10 (Figs. 6, S2). Furthermore, the results of RT-qPCR suggested higher expression of BCKDK and UAP1L1 in TMZ resistant glioma tissues than in TMZ sensitive glioma tissues (Fig. S3). The aforementioned results verified that upregulation of these target gene possibly mediated the TMZ resistance.

## DISCUSSION

Neoplasm remains the main killer worldwide (*Li et al., 2023*; *Li et al., 2018*; *Ma et al., 2016*; *Sun et al., 2021*; *Wu et al., 2018*; *Wu et al., 2017*; *Yang et al., 2019b*; *Zhang et al., 2023*). TMZ is the preferred alkylating agent for the treatment of GBM, and it induces cell death through DNA damage. Most patients develop TMZ resistance during treatment. Activation of the DNA repair pathway is the main mechanism underlying this phenomenon. With the deepening of research, newly proposed mechanisms of drug resistance, such as the regulation of microRNAs, membrane transporters, gap junction activity, and autophagy, still cannot solve this issue. There is a lack of effective treatments to maintain the effects of TMZ. In the present study, we screened and analyzed the differentially expressed proteins using UPLC-MS differential proteomics in two TMZ-resistant GBM cell lines, namely U251-TMZ and U87-TMZ. Compared with the U251 and U87 cell lines, 95 differential proteins were screened in the U251-TMZ and U87-TMZ cell lines, of which 28 proteins were upregulated and 67 proteins were down-regulated. The GO analysis demonstrated that the development of TMZ drug resistance may be related to intracellular immune regulation and energy metabolism of leukocytes. The KEGG analysis showed that the enriched pathways were related to energy metabolism. These preliminary data indicated the key role of energy metabolism in the development of TMZ resistance in GBM cell lines (*Fu et al., 2021*; *Manni & Min, 2022*; *Mao et al., 2022*). Through manual screening, we further identified seven proteins as potential GBM markers of TMZ resistance. Following this investigation and analysis of existing studies, we propose for the first time that UAP1L1 and BCKDK are potential markers of TMZ resistance in GBM.

UAP1L1 is a collateral homologue of UAP1 that participates in protein glycosylation and promotes the proliferation of some tumors. UAP1L1 is significantly upregulated in hepatocellular carcinoma tissues, and a high level of UAP1L1 expression is associated with poor prognosis (*Lai et al., 2019*). Recent studies have found that the expression of UAP1L1 is significantly upregulated in glioma tissues. Increased expression of UAP1L1 is associated with higher tumor grade and poor prognosis. UAP1L1 may promote the

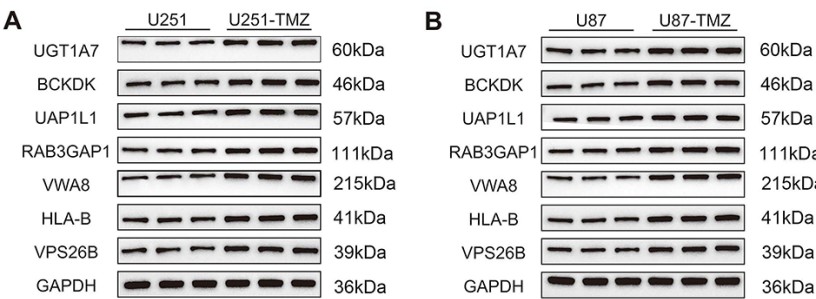

Fig. 5. Western Blotting showing the levels of upregulated proteins in U251/U251-TMZ and U87/U87-TMZ

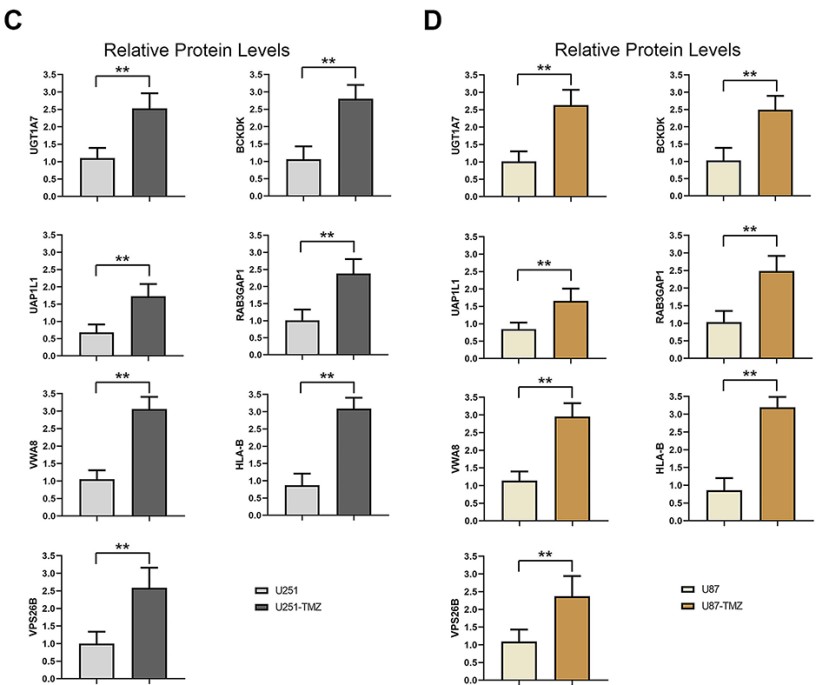

**Figure 5  Validation protein level expression of these upregulated genes by western blots.** (A) Protein expression of UGT1A7, BCKDK, UAP1L1, RAB3GAP1, VWA8, HLA-B and VPS26B in U251 and U251-TMZ cells. (B) Protein expression of UGT1A7, BCKDK, UAP1L1, RAB3GAP1, VWA8, HLA-B and VPS26B in U87 and U87-TMZ cells. (C) Quantification of three replicated data of each gene normalized to GAPDH from (A). (D) Quantification of three replicated data of each gene normalized to GAPDH from (B).

proliferation of glioma cells by regulating glycosylation of key proteins (*Yang et al., 2021b*). However, the specific mechanism by which UAP1L1 regulates glioma cell proliferation, apoptosis, and TMZ drug resistance is still poorly understood. BCKDK is a key regulator of branched-chain ketoacid dehydrogenase complex activity, and inactivates branched-chain ketoacid dehydrogenase through phosphorylation. Recent studies have shown that BCKDK regulates the catabolism of branched chain amino acids (BCAA) and enhances MEK/ERK signal transduction, which is the key pathway that drives the growth and proliferation of cancer cells (*Xue et al., 2017*). In patients with colorectal cancer, high BCKDK expression

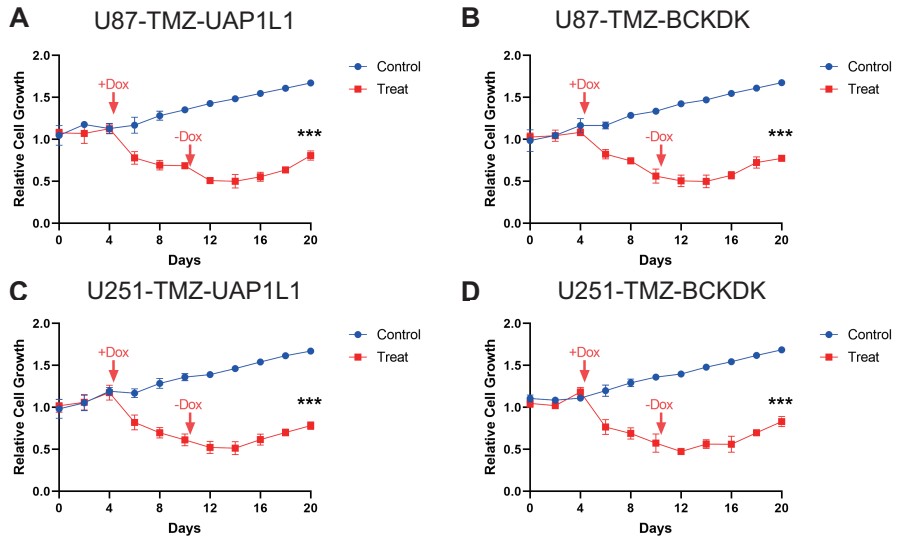

**Figure 6** **Verification of potential markers of TMZ resistance through knockdown of upregulated genes.** The doxycycline were added at Day 4 to reduce the expression of these four potential target gene, BCKDK (A), UAP1L (B) in U87-TMZ cells and BCKDK (C), UAP1L (D) in U251-TMZ cells and removed at Day 10.

is associated with poor prognosis (*Wang et al., 2021*). In addition, BCKDK can alter the metabolism of non-small cell lung cancer (*Wang et al., 2021*).

Owing to time constraints, we found only these two possible markers. GBM is one of the most difficult cancers to overcome in solid tumors. Photothermal therapy (PTT), photodynamic therapy (PDT), and sonodynamic therapy (SDT) are used in the field of oncology treatment as minimally invasive methods, as well as immunotherapies that are considered the most promising treatments. These treatments are very dependent on the use of active drugs, which are often designed for delivery into tumors. In addition, novel peptide-based molecular probes and multifunctional nanomaterials are used as different biomarkers and applied to tumor imaging during treatment. Therefore, nanomaterials based on self-assembled peptides have great potential to effectively improve the therapeutic effect on tumors by maximizing immunotherapy and non-invasive treatments (*Guo et al., 2022*; *Liu et al., 2022*; *Ma et al., 2020*; *Ren et al., 2020*; *Yang et al., 2021a*). In a follow-up study, it will be necessary to conduct an in-depth study of BCKDK and UAP1L1 based on the mechanism of TMZ resistance in GBM. This will be of great significance for the diagnosis, development, and application of TMZ resistance in GBM in the future.

## CONCLUSIONS

In this study, we established TMZ drug-resistant cell lines, U251-TMZ and U87-TMZ, in GBM. Through the screening and analysis of differential proteins using UPLC-MS differential proteomics technology, 95 differential proteins were screened using the U251-TMZ and U87-TMZ cell lines, of which 28 proteins were upregulated and 67 proteins were down-regulated. The co-upregulated proteins were studied using GO and KEGG analyses.

Most of the co-upregulated differentially expressed proteins were located in the cytoplasm and were significantly upregulated in biological processes related to endocytosis and myeloid leukocyte-mediated immune response, and may be involved in vesicle transport in the intimal system. These findings suggest that TMZ resistance may be related to immune regulation of leukocytes and intimal system transport. Subsequently, we screened and identified seven potential candidates for TMZ resistance. Combined with existing research, we propose for the first time that UAP1L1 and BCKDK are promising potential markers of TMZ resistance in GBM.

### Funding
The authors received no funding for this work.

### Competing Interests
The authors declare there are no competing interests.

### Author Contributions
- Da Li performed the experiments, analyzed the data, prepared figures and/or tables, authored or reviewed drafts of the article, and approved the final draft.
- Jun Yan performed the experiments, prepared figures and/or tables, and approved the final draft.
- Kang Li performed the experiments, prepared figures and/or tables, and approved the final draft.
- Qingcheng Yang analyzed the data, authored or reviewed drafts of the article, and approved the final draft.
- Liping Bian analyzed the data, authored or reviewed drafts of the article, and approved the final draft.
- Bencheng Lin analyzed the data, prepared figures and/or tables, and approved the final draft.
- Xiaohua Liu conceived and designed the experiments, authored or reviewed drafts of the article, and approved the final draft.
- Zhuge Xi conceived and designed the experiments, prepared figures and/or tables, and approved the final draft.

### Data Availability
  The raw measurements are available in the Supplemental Files.

### Supplemental Information
Supplemental information for this article can be found online at http://dx.doi.org/10.7717/peerj.16426#supplemental-information.

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
