# Peer review of "Identification of potential glioma drug resistance target proteins based on ultra-performance liquid chromatography-mass spectrometry differential proteomics"

_PeerJ, doi:10.7717/peerj.16426_

## Round 0.1 · original submission · Major Revisions

Based on the three review reports and my own evaluation, The current manuscript needs to be revised substantially.

Reviewer 1 ·

Basic reporting

The manuscript by Da Li involves the study of the “Identification of potential glioma drug resistance target proteins based on ultra-performance liquid chromatography-mass spectrometry differential proteomics”. Here, the author has used ultra-performance liquid chromatography-mass spectrometry (UPLC-MS) differential proteomics to screen for candidate markers of Temozolomide (TMZ) resistance in glioblastoma. This work has been performed very well with relevant methods. The results and discussion are well presented, and the text has been carefully written. But the author needs to improve the manuscript with some comments.
Comment:
1. In the material and method section lines 77-78 are not clear.
2. In the result section 1 lines 123-124 are not clear.

Experimental design

Experiments are very well designed and written.
With

1. Toxicity curve (Viability curve) for drug resistance and no of day to make drug resistance cells.
2. At what concentration do you start the treatment and what is the exact concentration for drug resistance? In the material and method section, you mention concentration increases up to 500µg/ml. is this dose is acceptable?
3. Is the dose continuous and alternate?
4. Figure 1b shows the effect on viability. What is the effective concentration for drug resistance.
5. Figure 2 shows the expression pattern. Is the analyzed data significant?
6. In figure 3b, selected molecules by the black circle are the top hit gene.
7. Figure 4 Please show the relation of signaling pathway and there significance in this work.
8. What is the novelty of the research article and the novel molecule for the drug resistance cells.

Validity of the findings

The experimental design and its validation are written very well.
But the author has some weak points in his article:

1. Author tries to explain the novel targets for glioma drug resistance, but they just represent the up-regulated gene, not the link with glioma.

Additional comments

NA

Reviewer 2 ·

Basic reporting

no comment

Experimental design

no comment

Validity of the findings

no comment

Additional comments

It is of great significance for the author to focus on the difficulty of tumor treatment. I find this paper is very interesting and identify the therapeutic target by high-throughput screening. This is a very meaningful screening. In this work, the authors established TMZ drug-resistant glioblastoma (GBM) cell lines, and screened differentially expressed proteins by UPLC-MS, and finally conducted data analysis. But the author needs to give a simple answer to the following question and make changes as suggested.

1. The title is "Identification of potential glioma drug resistance target proteins", while GBM is mainly introduced in the paper. And I think the paper should talk more about potential candidate targets.
2. Why does the author mention "UAP1L1 and BCKDK are promising potential markers of TMZ resistance in GBM." Because the purpose of the paper is to screen potential and differential proteins of TMZ resistance in GBM. Whether these proteins are indicators of TMZ-drug resistance needs to be verified by glioma clinical cohort.
3. In addition, the author mentions that "In this study, two classic GBM cell lines, U251 and U87, were used to identify reliable biomarkers for TMZ resistance ".
4. I believe that it should be a potential target for drug resistance, and these genes should be verified in follow-up experiments if the author had experimental conditions.
5. In Figure 4B, the author enriched many signaling pathways, but the relationship between these signaling pathways and drug resistance needs to be discussed. Refer to these documents ( PMID: 36226253, PMID: 34766135, PMID: 36276925).
6. The current progress of tumor treatment should be explained in the discussion section, please refer to citation (https://doi.org/10.1002/VIW.20200042; https://doi.org/10.1002/VIW.20200174; https://doi.org/10.1002/VIW.20200020,https://doi.org/10.1002/VIW.20200147; https://doi.org/10.1002/VIW.20220052).

Reviewer 3 ·

Basic reporting

The submitted manuscript (#81691) on, “Identification of potential glioma drug resistance target proteins based on ultra-performance liquid chromatography-mass spectrometry differential proteomics” reports the study conducted on U251 and U87 cell lines to screen for candidate markers of Temozolomide (TMZ) resistance in glioblastoma using UPLC-MS approach. The study observed 95 differentially regulated proteins in TMZ resistant cell lines as compared to control cell lines (28 proteins were upregulated and 67 proteins were down-regulated) of which UAP1L1 and BCKDK were proposed as promising potential markers of TMZ resistance among the seven identified candidate markers from GO and KEGG pathway analysis.
The manuscript is written well with clear background of the problem addressed, study objective and rationale. The language and grammar is good. Supportive references are given and format of bibliography is ok. Raw data along with figures and graphs are sufficient.

Experimental design

The study design is simple and the experimental procedures are as per the standards. The methodology as well as presentation of the results is straight forward with graphs, representative pictures, protein expression data and enriched interactive networks. The details about quality check of cell lines and ethical consent of the organization are not mentioned.

Validity of the findings

However the manuscript needs revision in view of the following concerns:
1. Of the identified seven candidate proteins- UGT1A7, BCKDK, UAP1L1, RAB3GAP1, VWA8, HLA-B, 208 and VPS26B, authors proposed UAP1L1 and BCKDK as promising markers for TMZ resistance. The western blot reveals differential expression for all the seven. The discussion with regard to this aspect is not clear. The authors may provide the justification.
2. The densitometry data is missing for western blots.
3. The results need to be validated functionally by checking in UAP1L1 and BCKDK downregulated cell lines. The authors are recommended to perform this step not only to confirm the observed results but also to delineate protein mediated pathological pathways in glioma through mechanistic studies.
4. Further, the expression status of the proposed markers could have been validated clinically by analysing in few patients samples.

Additional comments

The manuscript cannot be accepted in current form and needs major revision addressing the raised comments.

---

## Round 0.2 · accepted · Accept

Based on the reviewers' comments, the authors have addressed the concerns in an appropriate way that might be accepted for publication in the journal.

Reviewer 2 ·

Basic reporting

The authors found that seven candidates were identified as potential GBM markers of TMZ resistance. Combined with existing research findings, our study supports that UAP1L1 and BCKDK are promising potential markers of TMZ resistance in GBM. This is important for further understanding the molecular mechanisms that drive the development and enhancement of TMZ resistance.

Experimental design

The study design is very sound

Validity of the findings

The results are real and reliable

Additional comments

I don't have any more questions

Reviewer 3 ·

Basic reporting

The authors have revised the manuscript addressing all of the comments raised by the reviewers. The responses in the rebuttal are convincing.

Experimental design

The required data has been furnished and results have been refined.
However, the aim of study is identification of potential targets for drug resistance in GBM and the only two candidates have been picked up based on (mostly on earlier literature).
The novelty or significance of how current design helped in selection of targets is missing. The list of top 10 candidates and discussion on their relevance /interactome would have added more value to the manuscript.

Validity of the findings

The method of assessment of top two candidates is not very satisfactory. However, the study has its own significance is emphasizing the importance of these targets as promising in the future validation studies.
The missing/required data has been furnished and even justified.
Overall, the manuscript looks improvised.

Additional comments

no comments